# Field physical model tests on the mechanism of river blocking by debris flow in the middle reaches of the Dadu River, Southwest China

**Zhi Song**[1,2,3], **Yunxin Zhan**[1], **Yanni Chen**[2], **Gang Fan**[4]*

1 College of Earth Science and Engineering, Shandong University of Science and Technology, Qingdao, Shandong, P. R. China, 2 School of Tourism and Urban-rural Planning, Chengdu University of Technology, Chengdu, Sichuan, P. R. China, 3 Chengdu Center of China Geological Survey, Chengdu, China, 4 College of Water Resource and Hydropower, Sichuan University, Chengdu, Sichuan, P. R. China

* fangang@scu.edu.cn

## Abstract

Debris flow is a typical natural disaster in the middle reaches of the Dadu River, Southwest China. Field physical model tests were conducted to reveal the mechanism of river blocking by debris flow in the middle reaches of the Dadu River. The dynamic processes of riHver blocking by debris flows were revealed, and based on which three typical river-blocking modes of debris flow are observed, i.e. thrust-type river blocking, mixed-flow-type river blocking and progressive river blocking. The test results showed that the material composition of debris flows plays an important role in the river-blocking mode, only the tests that adopted the mixed soil and gravel exhibited the thrust-type river blocking mode. The material composition has a controlling effect on the thrust-type river-blocking model. Mixed-flow-type river-blocking mode appears most often in the tests with an intersection angle of 60˚, because the small intersection angle is conducive to the mixing of the debris flow and the water in the main channel. The debris flows composed of sand tend to block the river with mixed-flow-type river-blocking mode, accounting for 50% of the occurrences in the model tests. The high flow rate and discharge in the main channel and the low flow rate and discharge in the branch channel are the key factors controlling the progressive river-blocking mode. The test results in this study can support the debris flow disaster prevention and mitigation in this area.

**Data Availability Statement:** All relevant data are within the paper.

**Funding:** This research was financially supported by the Sichuan Province International Science and

## 1 Introduction

A debris flow is a typical natural disaster in mountainous areas; it is a torrent that carries a large amount of solid materials, such as sediment and stones. A debris flow is characterized by a suddenness, fast flow rate, large flow and strong destructive force. Such events can wash away transportation facilities and even villages and towns, causing huge losses. Debris flow disasters often last only a few hours and can even be as short as a few minutes. Historically, debris flow disasters have caused great losses around the world. Such as, in May 1970, a debris

Technology Innovation Cooperation/ Hong Kong, Macao and Taiwan Science and Technology Innovation Cooperation Project (grant number 2021YFH0178, received by Gang Fan), the Open Fund of State Key Laboratory of Geohazard Prevention and Geoenvironment (No. SKLGP2021K008, received by Gang Fan). The funders had no role in study design, data collection and analysis, decision to publish, or preparation of the manuscript.

**Competing interests:** The authors have declared that no competing interests exist.

flow disaster broke out in the town of Yange, Peru, with a flow velocity of nearly 100 m/s, about $5,000 \times 10^4$ m$^3$ of solid materials were washed out, causing nearly 20,000 deaths [1].

China has a complex topography with mountainous areas covering 66% of the land and has experienced some of the most serious debris flow disasters worldwide [2]. If the debris flows rush into the river, it will block the river and form a barrier lake and induce secondary disasters, the secondary disasters may cause more serious losses than the debris flow itself. Hence, the river-blocking by debris flows has attracted the attention of many researchers. The current study on the river-blocking by debris flows mainly focus on the following aspects: (1) tracking and investigating loose materials in a watershed using dynamic geological surveys [3]; (2) through data collection and comparison, to study the development characteristics of the basin and determine the differences in the background conditions of the river-blocking debris flows and the nonblocking debris flows [4]; (3) studying the characteristics of debris flow activities and obtaining the main control factors of the single ditch of the river-blocking debris flow [5]; (4) using experimental methods to analyze the rheological characteristics of a debris flow, obtain the formation age and scale, and analyze the possibility of river blocking by a debris flow formed in a single ditch [6]; and (5) adopting numerical methods to simulate the transition process of debris flow materials in a watershed and reproduce the sedimentation process of debris flow [2, 7].

River blocking by debris flows is an interaction between the debris flow and the water flow in the river channel [8, 9]. Researchers have revealed the mechanism of river blocking by debris flows through a variety of research methods. The existing studies mainly focused on the water flow velocity and water flow field at the intersection of the main river and the debris flows in tributaries [10–12], parameter sensitivity analysis [13–15], and scouring and sediment process analysis [16, 17]. Most of these studies are based on generalized models and do not focus on the actual rivers. The research results need to be validated in real conditions.

At present, there are few reports on the early identification of river blocking risk by debris flows, and most of the existing studies merely focus on the early identification of debris flow disasters using on-site investigation, remote sensing and GIS (Geographic Information System) technology, as well as identification modeling based on statistics. Among them, the identification of debris flows based on field investigations mainly considers ground investigations of the traces of accumulation fans and floodplain accumulations formed by early debris flows [18]. The recognition accuracy of this method is high, but the time and labor costs required are relatively large. Debris flow identification based on remote sensing and GIS technology and constructing relevant visual interpretation marks is an effective method [19]. Additionally, topography-related parameters such as the Melton index [20], roughness index [21], aspect ratio [22], sediment connectivity index [23], circularity ratio [24], elongation rate [25] and other coefficients are used to distinguish debris flow processes and river sediment transport processes. The discriminant model method is mainly based on the terrain parameters of the early debris flow basin [26, 27]. This method is conducive to the identification of debris flow trenches in a large area, but the identification results are greatly affected by the data accuracy and parameter selection. Studies on the prediction and early warning of debris flow disasters have been carried out. Early identification and early warning of debris flow disasters are mostly qualitative or semiquantitative methods based on the statistics of debris flow causative factors at the regional scale. In the selection of specific indicators, the topography of the watershed and debris flow excitation factors, such as precipitation, earthquakes and other factors, are generally considered [28–32]. With the development of computer technology, WebGIS technology is widely adopted in the monitoring and early warning of debris flow disasters, it has been adopted to provide early warning of debris flow disasters widely [33–35].

The existing studies and practices show that the acquisition of physical and mechanical parameters and motion characteristic parameters of debris flow is still a difficult problem, and it is difficult to obtain accurate physical and mechanical parameters and motion characteristic parameters of debris flow in practice. The difficult in parameter acquisition makes the research on the short-term prediction of debris flows and river blocking is still in its infancy, there is no comprehensive prediction indicator in current stage. Hence, field model tests are adopted in this study to reveal the mechanism of river blocking by debris flow, and it is expected to provide comprehensive prediction indicators for early warning of river blocking by debris flow eventually, and provide scientific support to the disaster prevention and mitigation in the middle section of the Dadu River.

## 2 Geological setting

The study area is the middle reaches of the Dadu River (Kangding-Shimian section), covering an area of 6,183.33 km$^2$. The deep valleys in the middle reaches of the Dadu River, coupled with strong tectonic activities and complex stratigraphic lithology, make this area a very active area for both internal and external dynamic geological effects and prone to frequent debris flow disasters [36–38]. The middle reaches of the Dadu River are illustrated in Fig 1. The complicated hydrodynamic conditions of the Dadu River cause strongly developed debris flows, which often block the Dadu River. Historically, many catastrophic debris flows have occurred in this area. For example, on July 23, 2009, the debris flow in Xiangshuigou, Kangding city, completely blocked the Dadu River, resulting in 16 deaths, 38 missing persons and considerable property losses [39]. On July 12, 2012, a debris flow in Tangjiagou, Shimian County, Ya'an city, blocked the Dadu River, resulting in 2 deaths and 5 missing persons [40]. On July 4, 2013, the Xiongjiagou debris flow completely blocked the Zhuma River, resulting in 19 deaths [41]. Other typical debris flow disasters in the study area are shown in Fig 1.

The study area is crisscrossed by water systems, with the development of tributaries such as the Moxi River, Wasigou, Nanya River, Anshun River and Tianwan Gou. The valleys are deep, and the terrain is undulating. The middle reaches of the Dadu River are located in the transition zone from the Sichuan Basin to the Qinghai-Tibet Plateau. The middle reaches of the Dadu River are divided into three sections, the Kangding section, Luding section and Shimian section. Affected by the climatic conditions of the Tibetan Plateau and monsoons, the middle reaches of the Dadu River undergo local heavy rainfall, and local heavy rainfall is an important external cause that directly triggers debris flows. There are a number of tributaries in this area, each tributary may have different confluence characteristics, as evidenced by the floods of each tributary not occurring at the same time, leading to the obvious "pulse" characteristics of the flood confluence in the study area. The average annual total flow in this section is 330.56×108 m$^3$, the average annual flow is 1,218 m$^3$/s, the maximum peak flood is 6,050 m$^3$/s.

The middle reaches of the Dadu River consist of towering mountains and deep valleys, and the relative elevation difference is approximately 5,200 m. The depth of the V-shaped valley is 700–1,400 m. The middle section of the main stream of the Dadu River develops along the Dadu River fault zone, which is a zone of abrupt changes in topography and landforms that constitutes the eastern boundary of the Qinghai-Tibet Plateau.

The middle reaches of the Dadu River are located on the eastern margin of the Qinghai-Tibet Plateau, and all other strata are exposed except for Cretaceous and Cambrian strata. Within the 6,183.33 km$^2$ study area, the strata with a distribution area greater than 1,000 km$^2$ are the Nanhua System, Proterozoic, and Archaeo-Proterozoic strata, which are relatively old strata. Among them, the Nanhua System strata are mostly distributed on the left bank of the lower Dadu River and are exposed in patches. The Proterozoic strata are distributed along the

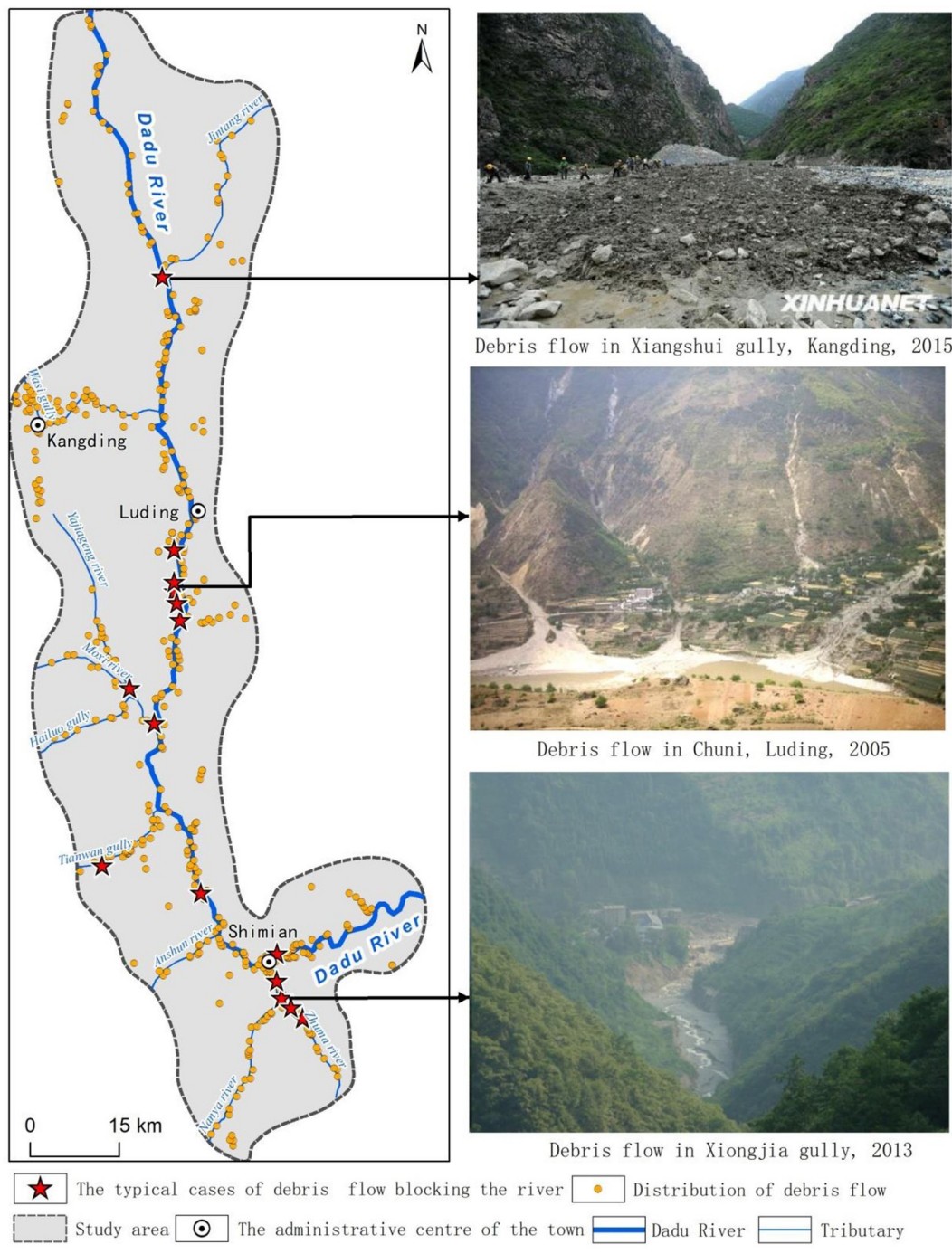

**Fig 1. Typical debris flow cases in the study area.**

Kangding-Luding line on the right bank of the upper Dadu River. The Archean-Proterozoic strata are distributed on the right bank of the upper Dadu River, and the distribution area of the above three strata accounts for 80% of the total area of the study area.

The middle reaches of the Dadu River are macroscopically located at the tectonic intersection with complex structures. The north side is the Sichuan-Qinghai block, the west side is the Sichuan-Yunnan block, and the east side is the South China block. The NW-trending

Xianshuihe fault, the NE-trending Longmenshan fault and the N–S-trending Anninghe fault constitute a Y-shaped structure in the Shimian section.

However, with the development of the economy, the vegetation in the study area is continuously destroyed, increasing the possibility of debris flow disasters. Meanwhile, the study area frequently experiences local heavy rainfall; therefore, this area is vulnerable to debris flow disasters.

## 3 Distribution of regional debris flows

There are a total of 441 debris flow ditches in the study area, which are divided into four categories in terms of scale: extra large, large, medium and small. Among them, there are 6 extra large debris flows, accounting for 1.36% of the total number of debris flows; 45 large debris flows, accounting for 10.20%; 224 medium debris flows, accounting for 50.79% of the total; and 166 small debris flows, accounting for 37.64% of the total. This indicates that in the Dadu River Basin, the debris flows are mainly small and medium in size, with relatively few extra large scale debris flows.

In terms of the geomorphological conditions for the formation of debris flows, there are 436 gully debris flows, accounting for 98.87% of all identified debris flows, and there are 5 slope debris flows, accounting for 1.23% of all identified debris flows, mainly located on the right bank of the Dadu River in Shimian County. The debris flows in the Dadu River Basin are strongly influenced by topography, mainly due to the high and steep slopes of the Dadu River Basin, which are not conducive to the accumulation of sources on the slope. In terms of the material composition of these debris flows, there are 358 true debris flows, accounting for 81.15% of all identified debris flows, followed by water–rock flows, a total of 81, accounting for 18.34% of all identified debris flows; there are only 2 mud flow cases, accounting for 0.51% of all identified debris flows. Residual slope deposits and abandoned mineral slag are the main sources of the debris flows in the study area.

The debris flows in the study area are mainly distributed along both sides of the main water system. The average distribution density of geological disasters is 0.14/km$^2$, the maximum density is 10.2/km$^2$, and debris flow disasters are mainly distributed in the Shimian section, as illustrated in Fig 2. This area is located at the intersection of the three major fault zones of the Anning River, Xianshui River and Longmen Mountain, with strong tectonic activity, abundant loose deposits, and heavy rainfall.

## 4 Physical model tests

In this study, field physical model tests were conducted to reveal the three blocking modes of debris flows, to study and analyze the influencing factors of river blocking, and to study the characteristics of the blocking process, providing a basis for the early identification of river-blocking modes in the study area.

### 4.1 Test devices

In this study, the confluence area of the Moxi River and Dadu River, which is one of the typical disaster zones in the study area, was selected as the prototype of these physical model tests. The Moxi River basin is located on the right bank of the Dadu River, with a drainage area of 923 km$^2$. The elevation difference in the Moxi River basin is generally 1,500~3,000 m, and colluvial deposits are abundant in the basin. In August 2005, the Moxi River basin experienced a large-scale debris flow disaster. After the debris flow entered the Dadu River, a considerable accumulation fan was formed, and the Dadu River was blocked for approximately 20 minutes.

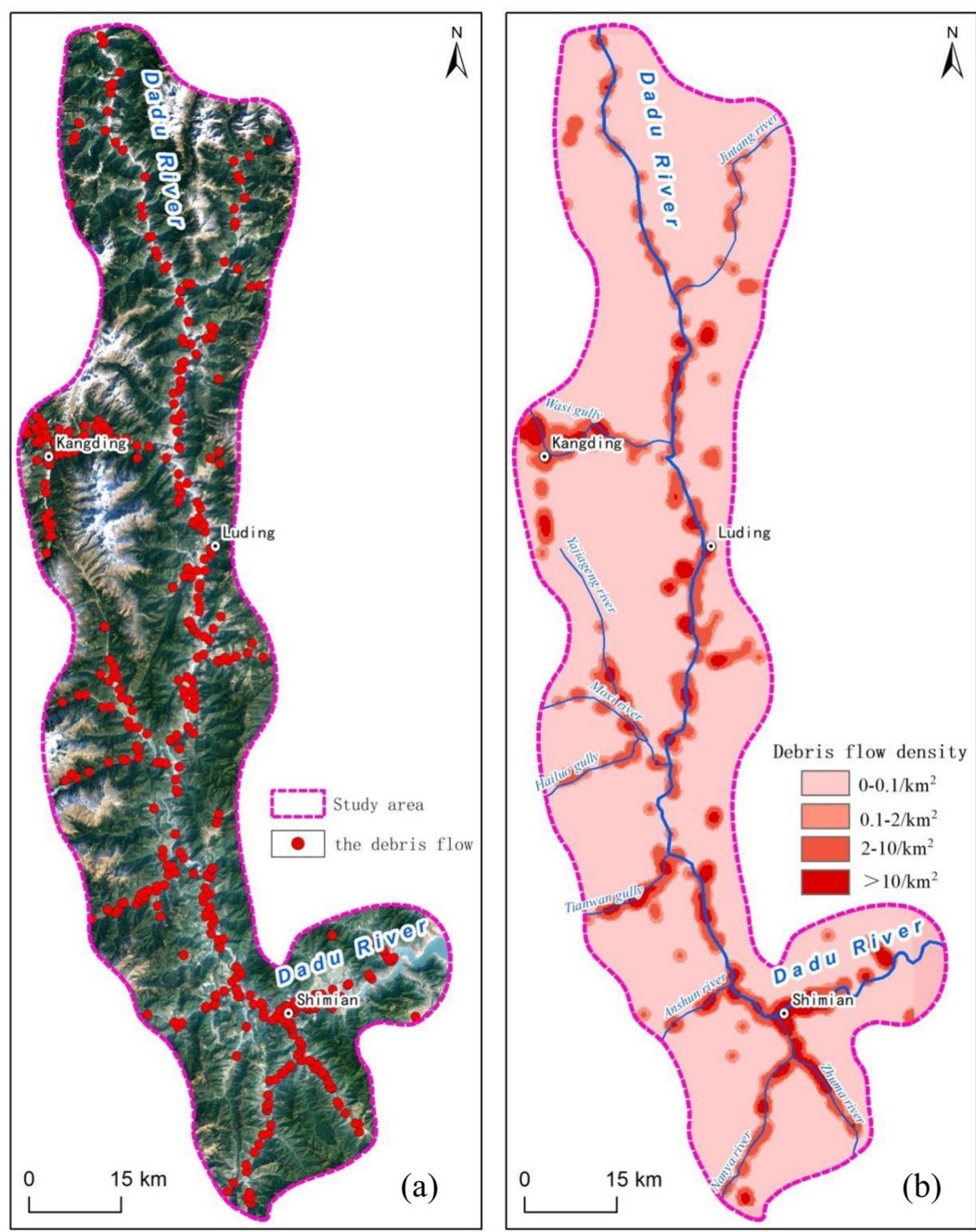

**Fig 2.** (a) Distribution of debris flows and (b) debris flow density in the study area.

The test model was completed on site in the study area. The test device was mainly composed of the storage tank, the waste tank, the water storage tank, the main channel and the branch channel, as illustrated in Fig 3. The three tanks were made of iron sheets. The bottom of the main channel and the branch channel was an iron sheet, while the sidewalls were organic glass, which was convenient for debris flow observation. In these tests, a triangular weir was used to manually control the flow velocity and flow discharge in the main channel, and SVR Dymeco surface radar was used to measure the flow velocity. A high-precision

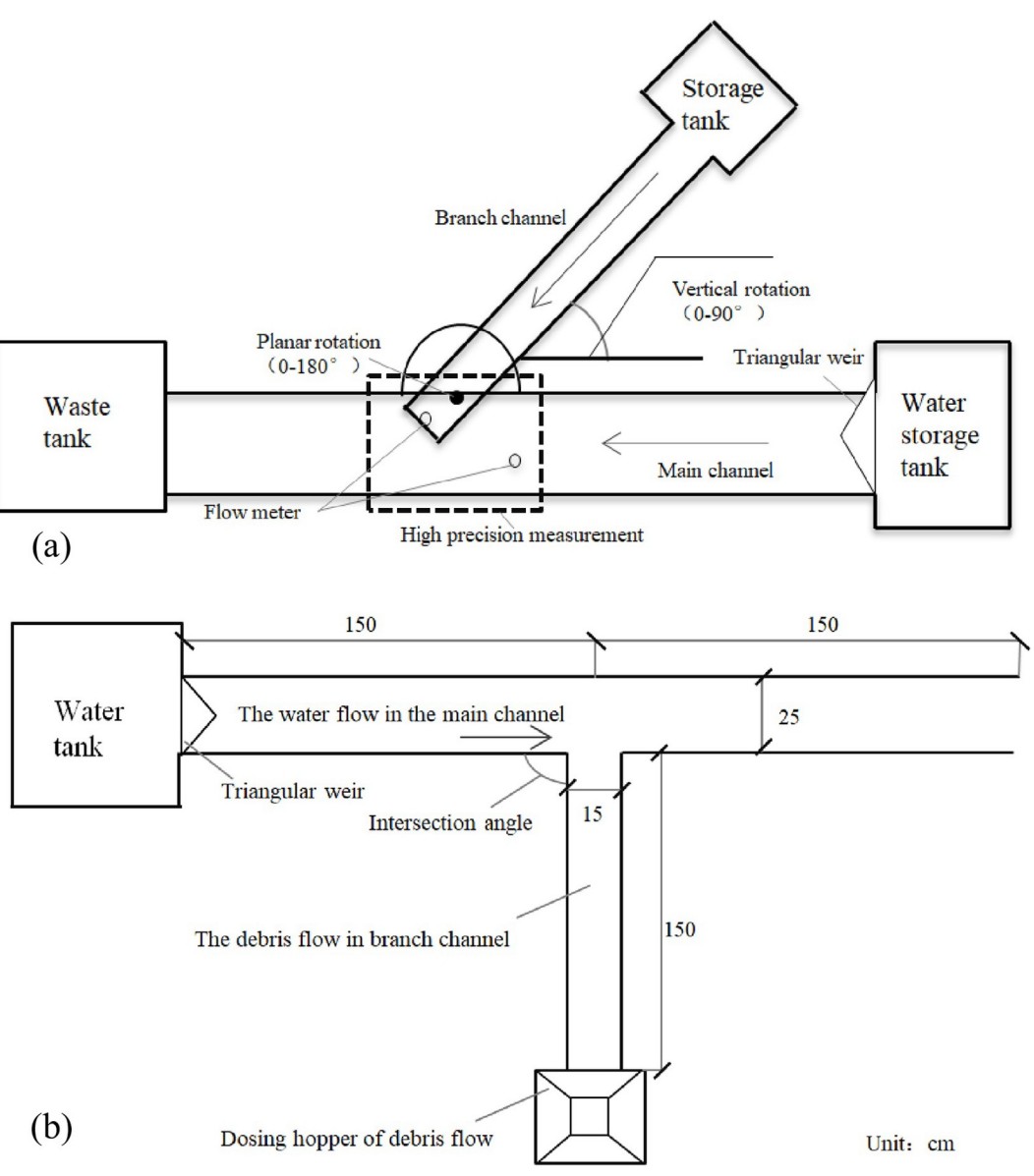

**Fig 3.** Schematic diagram of the test device: (a) flat view, (b) top view.

camera was utilized to record the morphology of the deposits at a speed of 1 picture/0.2 seconds, and the pictures obtained at the test site were converted into obtain sedimentation distribution maps with CAD software. The inflow flow of the branch channel was converted from the measured average flow velocity and cross-sectional area at the intersection. Recording rulers were placed at the intersection to measure the width and height of the mud and location of debris flow movement.

The width of the main channel was 20 cm, the width of the branch channel was 15 cm. The test device was movable, the branch channel was slightly higher than the main channel by 5 cm, and the angle in the vertical direction could be adjusted between 0° and 90° to simulate the longitudinal slope of the main channel of the natural debris flow channel. The intersection angle between the branch channel and the main channel on the plane could be adjusted

between 0˚ and 180˚ to simulate the intersection angle between the debris flow channel and the river, as illustrated in Fig 3.

The similarity ratio is the key to physical model testing. The confluence zone of the Moxi River and the Dadu River was selected as the prototype, and a variety of variables were involved in this study. According to the limitations of the field test site and the capacity of the test equipment, the geometric scale of this model test was finally determined to be 1:500. The similarity ratios adopted in this study are shown in Table 1.

## 4.2 Test conditions

Based on the investigation of the intersection angle in the study area, the intersection angles in the physical model tests were determined to be 60˚, 90˚ and 120˚. Herein, the intersection angle refers to the angle between the debris flow direction in the branch channel and the direction of the main river. The average slope of the riverbed in the Dadu River Basin varies in the range of 3–9˚; hence, the longitudinal slope of the main channel was set to 4˚ and 8˚. The longitudinal slope of the debris flow branch channel in the study area is 4–16˚, and the slope of the branch channel was set to 5˚, 10˚ and 15˚ in the model tests. The stratum in the study area is complex. Considering the feasibility of the field model test, five materials were selected for the test, namely, clay, silt, sand, gravel, and mixed soil, as shown in Fig 4. All the test materials were collected from the debris flow accumulation fan in the study area to simulate the prototype material. The densities of the different types of materials were measured and are shown in Table 2.

Considering the intersection angle, slope of the main channel, slope of the branch channel and test material, a total of 90 groups of physical tests were carried out in this study. The control range of the debris flow density in this test was 1,600–2,830 kg/m$^3$. The flow rate in the main channel was manually controlled by the triangular weir, the flow rates were 0.5 m/s and 0.6 m/s, and the discharges were 1.5 m$^3$/s and 1.8 m$^3$/s, respectively. The measurement results showed that the flow velocity in the branch channel was 0.8–1.6 m/s, and the flow discharge was 2.4–4.81.8 m$^3$/s. Herein, the flow velocity ratio is defined as the ratio of the flow velocity in the branch channel to the flow velocity in the main channel, and the discharge ratio is defined as the ratio of the flow discharge in the branch channel to the flow discharge in the main channel.

## 4.3 Dynamic process of river blocking

The following types of dynamic processes of river blocking by debris flow were observed in physical model tests.

(1) When the flows in the branch channel and main channel were relatively large and the water depth of the main river was not deep, the debris flow in the branch channel entered the main channel instantaneously. Then, the water level of the main channel rose and formed backwater in the main channel. As the water level in the main channel continued to rise, a dammed lake was formed in the main channel, the dam gradually began to leak, and finally, the dam breached.

**Table 1. Similarity ratios of the tests.**

| Parameter | Symbol | Similarity ratio | Parameter | Symbol | Similarity ratio |
|---|---|---|---|---|---|
| Geometric length | $\lambda_l$ | 500 | Density | $\lambda_\gamma$ | 1 |
| Velocity | $\lambda_\mu$ | $10\sqrt{5}$ | Void ratio | $\lambda_e$ | 1 |
| Discharge | $\lambda_Q$ | $500^{2.5}$ | Grading | $\lambda_{pi}$ | 1 |
| Time | $\lambda_t$ | $10\sqrt{5}$ | Grain size | $\lambda_d$ | 500 |

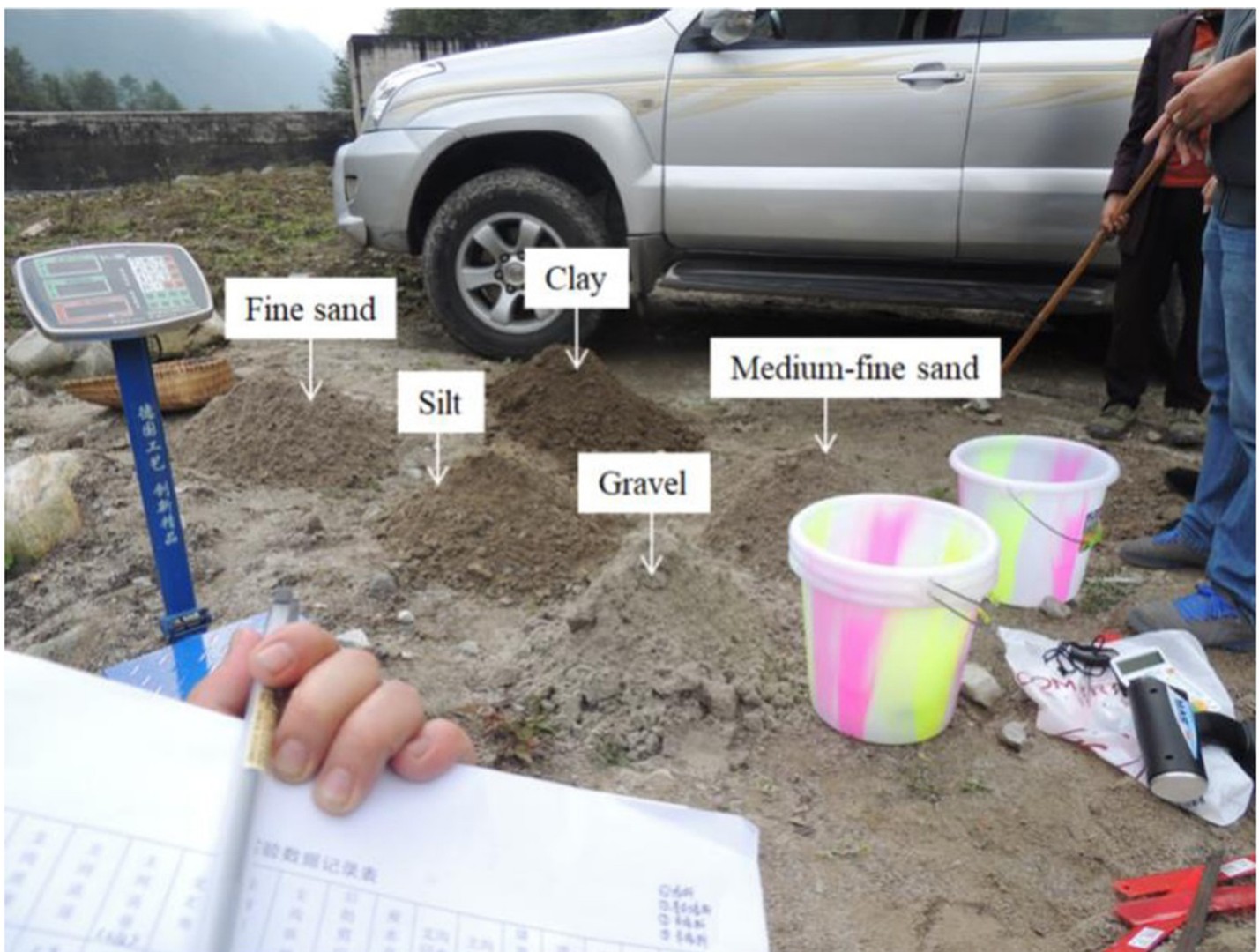

**Fig 4. Different types of material adopted in this physical model test.**

(2) Another typical dynamic process of the debris flows was that when there were many solid components in the debris flow, some of the solid components flowed into the main river, moved to the opposite bank and deposited, and some smaller particles in the debris flow were mixed with the main river flow, causing a local increase in the sediment content and

**Table 2. Density ranges for different types of materials.**

| Type | Measured density(kg/m$^3$) |
|---|---|
| Clay | 1,600–1,930 |
| Silt | 1,680–2,190 |
| Sand | 1,780–2,320 |
| Gravel | 1,890–2,600 |
| Mixed soil | 1,980–2,830 |

forming a high sediment flow. Finally, the main channel was blocked, a dammed lake formed, and the dammed lake breached due to the water level increase.

(3) Third, after the debris flow entered the main channel, the debris flow head interacted with the water flow in the main channel, and finally, the debris flow head reached downstream of the main channel, resulting in a narrow cross-section of the main channel and causing the local flow in the main channel to increase instantaneously, with local deposition. Lateral moraines were formed on both sides downstream of the main channel. This kind of dynamic process of river blocking mainly occurs when the water flow on both sides is slower than the water flow in the middle of the main channel, the water depth is shallower, and there are fewer large particles in the debris flow.

## 4.4 Analysis of river-blocking mode

To quantitatively describe the influence of river blocking by a debris flow, this study defines the maximum blocking degree $S$ as the ratio of the maximum width of the blocking body (temporary or permanent) formed by debris flow $a$ to the actual width of the main river, which can be used as a quantitative characterization index for river blocking $L$, i.e., $S = a/L$. According to the on-site physical model test, the river-blocking modes of debris flow can be divided into three types, namely, thrust-type river blocking, mixed-flow-type river blocking and progressive river blocking.

**4.4.1 Thrust-type river blocking.** The debris flow enters the main channel quickly, completely blocking the main channel and forming a typical barrier dam. The process of river blocking can be divided into the following stages: (1) the debris flow thrusts into the main channel instantaneously, (2) the head of the debris flow rushes to the opposite bank, (3) the source material of the debris flow fills the main channel, (4) the water level of the main channel rises, (5) backwater forms in the main channel and branch channel, (6) seepage occurs in the weak zone of the barrier dam, (7) local collapse occurs in the barrier dam, and (8) finally, the barrier dam breaches. This river-blocking mode mainly occurs when the debris flow density is high, the debris flow is mainly viscous, the shear stress resistance of the debris flow material is strong, the discharge ratio is large and the water depth in the main channel is small. The main reason for this river-blocking mode is that when the total amount of solid materials washed out by the debris flow is large and the deposition rate is fast, the discharge is relatively high, and the water flow in the main river cannot scour away the accumulated debris flow material in time, nor can it change the flow direction to make the debris flow move downstream; hence, the incoming debris flow materials will move to the opposite bank of the main river quickly, and the river will become blocked by a temporary natural dam, as illustrated in Fig 5.

The river blocking phenomenon was observed in 70 groups of tests, and the river was completely blocked in 22 groups of tests, accounting for 24.44% of the total number of tests. The maximum blocking degree $S$ is 1 in this river-blocking mode, and the maximum length of the barrier dam is 20 cm. The occurrence times for each river-blocking mode under the different test conditions are listed in Table 3.

Only the tests that adopted the mixed soil and gravel exhibited this kind of river-blocking mode. The material composition of debris flows plays an important role in the river-blocking mode. The measured density of the tested debris flows is high, ranging from 2,100–2,830 kg/m$^3$. The thrust-type river-blocking mode appears 9 times in the tests with a flow velocity of 0.6 m/s and 13 times in the tests with a flow velocity of 0.5 m/s. This kind of river-blocking mode occurs 9 times under conditions with a flow rate of 1.5 L/s in the main channel and 13 times in the tests with a flow rate of 1.8 L/s in the main channel. The flow rate in the branch channel is

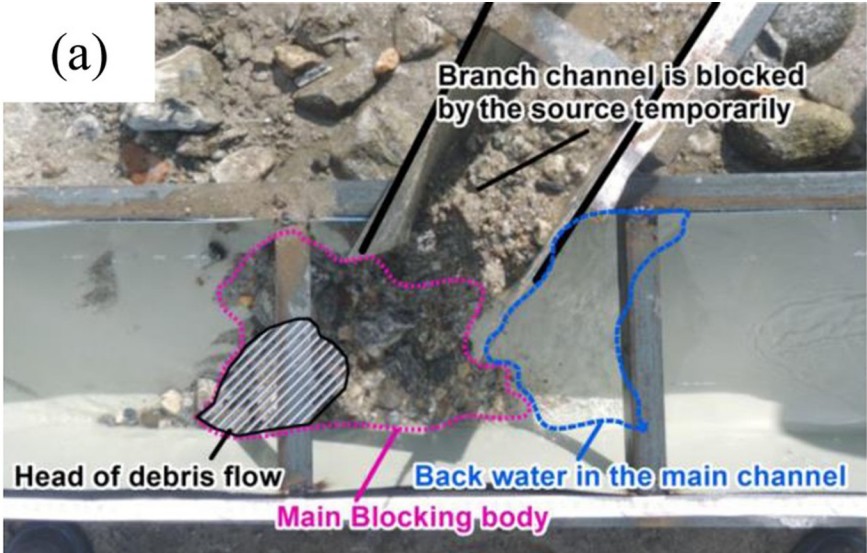

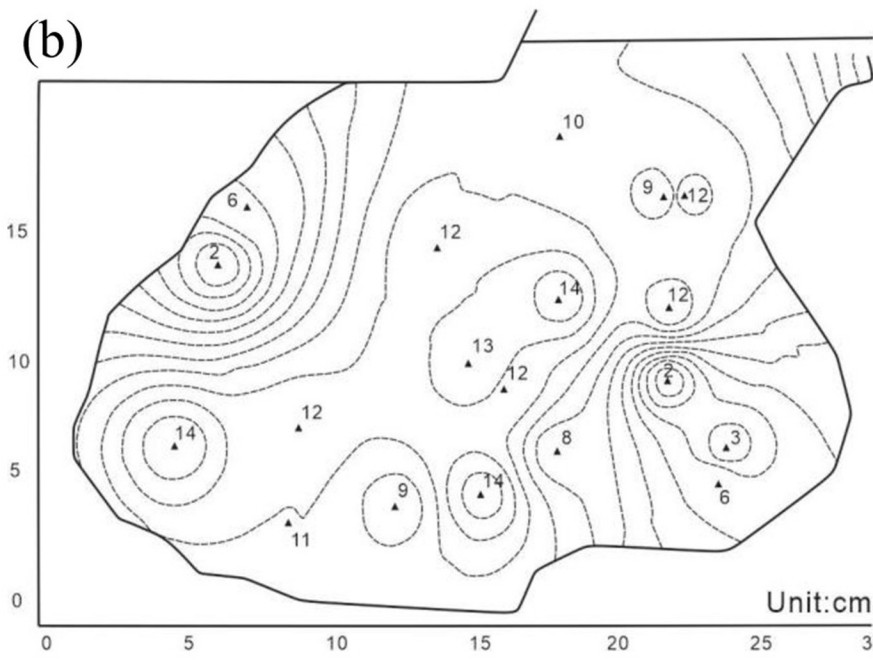

**Fig 5.** Illustrations of the thrust-type river-blocking mode, (a) test result and (b) the sedimentation form.

**Table 3. Occurrence times for each river-blocking mode under different test conditions.**

| River blocking mode \ Test conditions | Total number of times the condition occurs | Intersection angle | | | Longitudinal slope of the main channel | | Longitudinal slope of the branch channel | | |
|---|---|---|---|---|---|---|---|---|---|
| | | 60° | 90° | 120° | 4° | 8° | 5° | 10° | 15° |
| Thrust-type river blocking | 22 | 5 | 7 | 10 | 9 | 13 | 5 | 8 | 9 |
| Mixed-flow-type river blocking | 21 | 9 | 6 | 6 | 12 | 11 | 7 | 7 | 7 |
| Progressive river blocking | 27 | 12 | 7 | 8 | 15 | 12 | 11 | 9 | 7 |

between 1.2 and 1.6 m/s, and the discharge of the branch channel is between 3.6 and 4.8 L/s. The flow velocity ratio and the discharge ratio are between 2.00 and 3.20. The height of the backwater is 13–20 cm, the length of the blockage body is 20 cm, the minimum dam breaching duration is 1.5 s, and the maximum value is 6.5 s. The observation results showed that approximately 700–1,600 cm$^3$ solid materials were washed away in the main channel, mainly in a range of 800–1,000 cm$^3$.

As mentioned above, the material composition of debris flows plays an important role in the river-blocking mode. By analyzing the relationship between the measured density of the debris flow materials and the degree of river blocking, the correlations between the measured density of the debris flow materials and the height of the backwater, the breaching duration of the barrier dam, and the amount of material that was washed away were obtained for the thrust-type river-blocking mode, as illustrated in Fig 6. Since the linear function is the optimal one with the maximum correlation coefficient, hence, the linear function is adopted in this study to illustrate the relationships between different parameters. Among them, the linear correlation coefficient of the measured density and backwater height reaches 0.9705, which is a strong positive correlation; the linear correlation coefficient of the measured density and breaching duration reaches 0.8891, while the linear correlation coefficient of the measured density and the amount of material that was washed away is 0.5593. The above correlation analysis indicates that the material composition has a controlling effect on the thrust-type river-blocking model.

**4.4.2 Mixed-flow-type river blocking.** In this river-blocking mode, the process of river blocking can be divided into the following stages: (1) when the debris flow enters the main river, it almost fills the main channel, (2) it forms an instantaneous blockage in the main channel, (3) the water flow and the debris flow materials mix near the barrier dam, (4) the water level rises, and (5) due to the flow action being strong at the intersection zone of the main and branch channels, the main body of the barrier dam collapses, and the dam breaches. The features of mixed-flow-type river blocking are characterized by a viscous debris flow with a high density, a large flow velocity ratio and strong flow action in the main channel. The main reason for the mixed-flow-type river-blocking mode is that part of the solid components of the debris flow are transported into the main river and move to the opposite bank, where they are deposited. Other fine particles in the debris flow mix with the main river flow. The debris flow is subjected to shear force and disturbance from the main river flow, and the mixing of the debris flow and the water flow causes an increase in the sediment content in local areas, forming a high sediment flow that rushes downstream, as illustrated in Fig 7.

This kind of river-blocking mode was observed in 21 groups of tests, accounting for 23.33% of the total number of tests. The tests adopting the mixed soil, gravel, sand and silt exhibited this kind of river-blocking mode. Among them, 10 tests included sand, 8 tests included gravel, 2 tests included mixed soil and 1 test included silt. This indicates that sand is more likely to result in mixed-flow-type river blocking.

The measured density of the tested debris flows is between 1910 and 2,560 kg/m$^3$. This kind of river-blocking mode appears 9 times in the tests with a flow velocity of 0.6 m/s and 12 times in the tests with a flow velocity of 0.5 m/s. It occurs 9 times in the tests with a flow rate of 1.5 L/s in the main channel and 12 times in the tests with a flow rate of 1.8 L/s in the main channel. The flow rate in the branch channel is between 0.8 and 1.6 m/s, and the discharge of the branch channel is between 2.4 and 4.8 L/s. The flow velocity ratio and the discharge ratio are between 1.6 and 2.8. The height of the backwater is 8–15 cm, the length of the blockage body is 12–15 cm, and the maximum dam breaching duration is 4 s. The observation results showed that approximately 700–3,000 cm$^3$ of solid materials were washed away in the main channel.

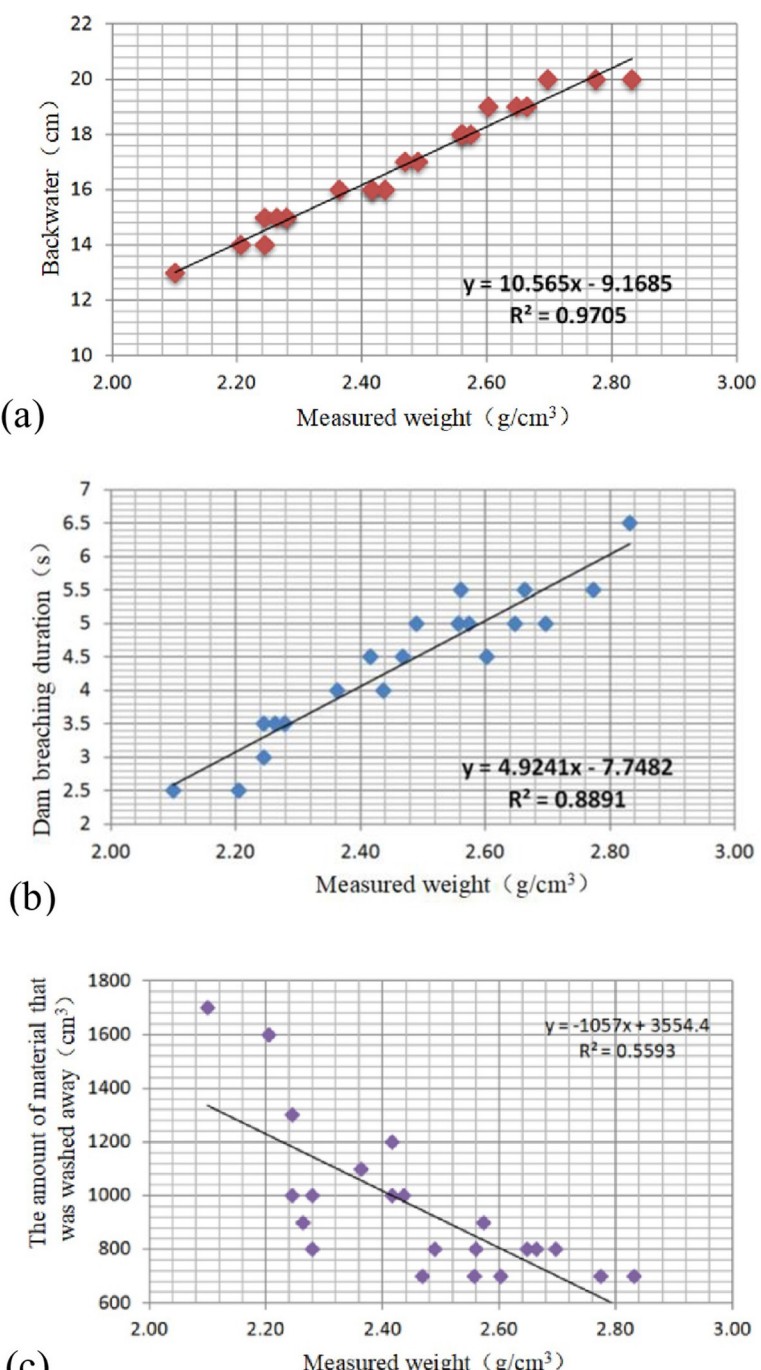

**Fig 6.** Correlation between measured density and (a) backwater height, (b) dam breaching duration and (c) the amount of material that was washed away.

In the mixed-flow-type river-blocking mode, the controlling effect of topographic conditions, i.e., the intersection angle and the longitudinal slopes of the main and branch channels, is weak. The statistics on the intersection angles show that this river-blocking mode appears most often in the tests with an intersection angle of 60° because the small intersection angle is

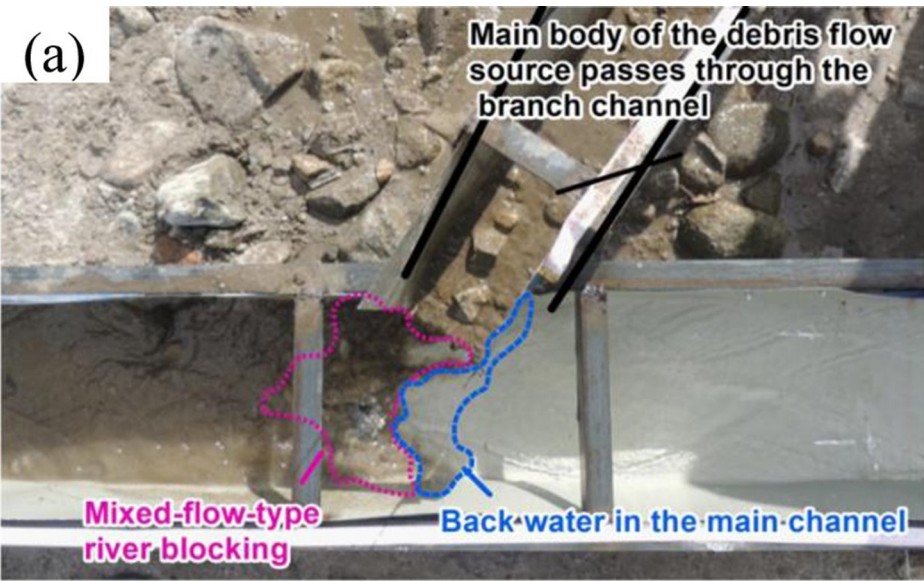

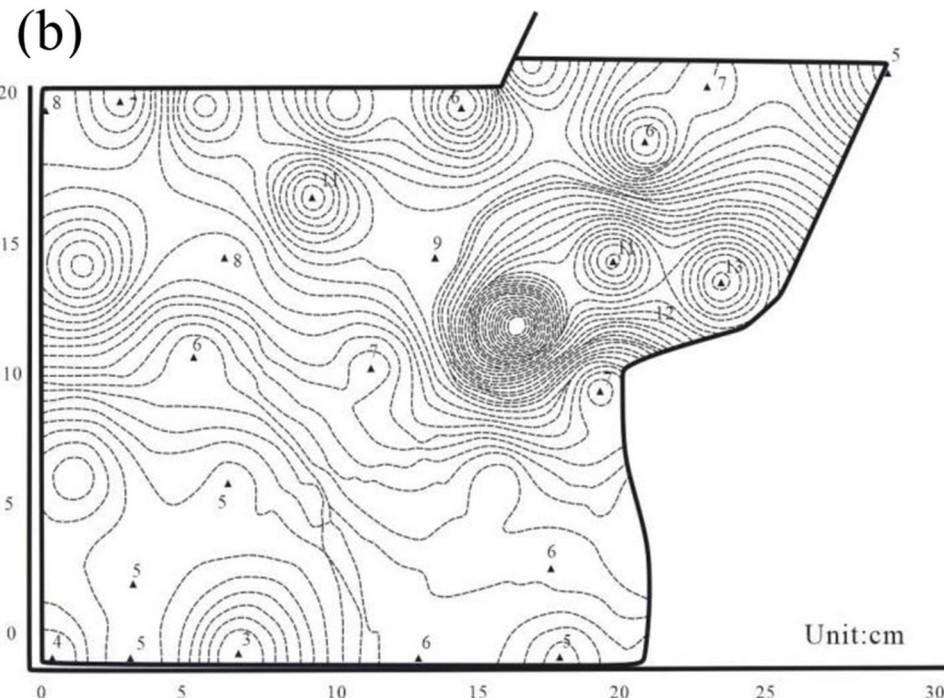

**Fig 7.** Illustrations of the mixed-flow-type river-blocking mode, (a) test result and (b) the sedimentation form.

conducive to the mixing of the debris flow and the water in the main channel. This is quite different from the thrust-type river-blocking mode.

The test results show that the longitudinal slope geomorphological conditions have a weak influence on river blocking by debris flow in this mode. In the mixed-flow-type river-blocking mode, the material compositions result in the following order of the number of mixed-flow-type river-blocking occurrences, from most to least: sand, gravel, mixed soil, and silt; among them, the debris flow composed of sand accounts for 50% of the occurrences. The maximum

blocking degree $S$ is 0.3–1.0 for sand, 0.5–0.9 for gravel, 0.75–0.85 for mixed soil, and only 0.25 for silt. Compared to those of other materials, the particle size and density of sand are more likely to block the river with this mode, and the maximum blocking degree $S$ can even reach 1.0.

According to the definition, the larger the flow velocity ratio and discharge ratio are, the higher the branch channel energy and the weaker the main river energy. Theoretically, the larger the ratio is, the more conducive it is to river blocking. In this mode, because the cross-sectional area was constant, the flow velocity ratio is the same as the discharge ratio; herein, only the discharge ratio was analyzed. The analysis shows that the discharge ratio has a high positive correlation with the maximum blocking degree $S$ and the height of backwater, and the linear correlation coefficients are 0.8864 and 0.8566, respectively. As the discharge ratio increases, the debris flow in the branch channel becomes greater and more active, and it is easy to block the main river and form a large-scale barrier dam. Meanwhile, the water height in the main river is more likely to rise, leading to an increase in the backwater height.

The results of the correlation analyses between the discharge ratio and the amount of wash-out and duration of dam breaching are shown in Fig 8. The discharge ratio is highly negatively correlated with the amount of washout, with a correlation coefficient of 0.8252. This is because in the mixed-flow-type river-blocking mode, the weaker the kinetic energy of the debris flow material flowing into the main river, the easier it is to be carried away by the main river, while the stronger the kinetic energy of the debris flow, the easier it is to form a barrier dam in the main river. The linear correlation coefficient for the dam breaching duration is only 0.1604, indicating that the intersection of the debris flow with the main river in the mixed-flow-type river-blocking mode is complex and random.

**4.4.3 Progressive river blocking.** The stages of river blocking in this mode are as follows: (1) the debris flow rushes into the main channel, (2) the head of the debris flow interacts with the flow in the main channel, (3) the head of the debris flow reaches downstream, (4) the section of the main channel becomes narrow, (5) local discharge increases instantaneously, (6) the main river is blocked progressively and (7) lateral moraines are formed on both down-stream sides. The progressive river-blocking mode mainly appears for the diluent debris flows with a strong flow in the main river. The main reason for this river-blocking mode is that the flow on both sides of the main channel is slower and shallower than the flow in the middle of the main channel, and the debris flow contains fewer large particles. The debris flow can reach the opposite bank after entering the main channel, but the debris flow materials in the middle part are washed away or deposited downstream. As the debris flow continues to enter the main channel, the deposits on both sides gradually widen toward the river center, and the nar-rowing of the cross-section leads to an increase in flow velocity; hence, the subsequent incom-ing debris flow material cannot be deposited here, and the river will not be blocked by the debris flow, as illustrated in Fig 9.

This kind of river-blocking mode was observed in 27 groups of tests, accounting for 30% of the total number of tests. The tests that result in this river-blocking mode included all the kinds of materials. Among them, 13 tests included silt, 8 tests included sand, 3 tests included gravel, 2 tests included clay and only 1 test included mixed soil. This indicates that sand is more likely to result in mixed-flow-type river blocking.

The measured density of the debris flow is between 1,680 and 2,320 kg/m$^3$, lower than the abovementioned two kinds of river-blocking modes. This kind of river-blocking mode appears 12 times in the tests with a flow velocity of 0.6 m/s and 15 times in the tests with a flow velocity of 0.5 m/s. This kind of river-blocking mode occurs 15 times in the tests with a flow rate of 1.5 L/s in the main channel and 12 times in the tests with a flow rate of 1.8 L/s in the main channel. The flow rate in the branch channel is between 0.8 and 1.4 m/s, and the discharge of the branch channel is between 2.4 and 4.2 L/s. The flow velocity ratio is between 1.6 and 2.6. The height of

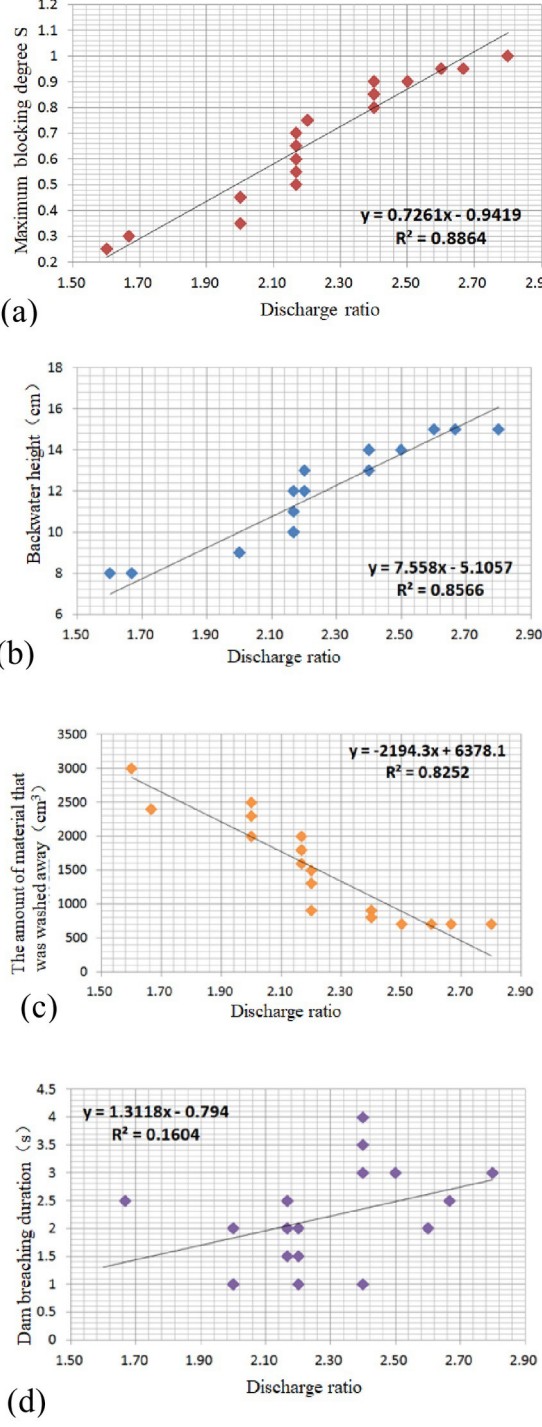

**Fig 8.** Correlation between discharge ratio and (a) the maximum blocking degree *S*, (b) backwater height, (c) the amount of material that was washed away and (d) dam breaching duration.

the backwater is 8–15 cm, and the length of the blockage body is 2–20 cm, mainly in the range of 8–10 cm. The dam breaching duration mainly ranges between 1 and 3.5 s. The observation results showed that approximately 800–3,800 cm$^3$ solid materials were washed away in the

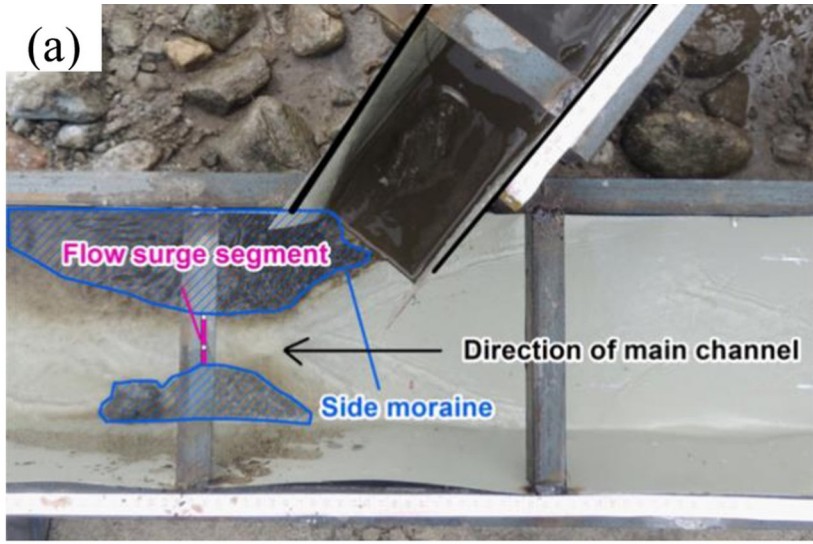

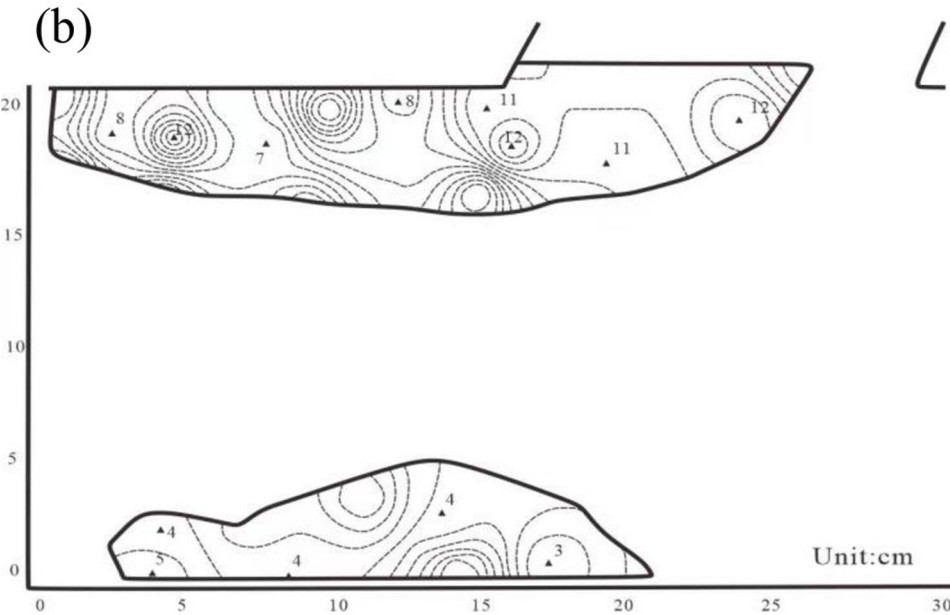

**Fig 9.** Illustrations of the progressive river-blocking mode, (a) test result and (b) the sedimentation form.

main channel, mainly in a range of 2,800–3,000 cm³. The test results indicate that the influence of geomorphological conditions is tiny in progressive river blocking. The correlation between the measured density and the maximum blocking degree $S$ is plotted in Fig 10.

The results of the correlation analyses between the discharge ratio and the backwater height, the maximum blocking degree $S$, duration of dam breaching, and the amount of washout are shown in Fig 11. The discharge ratio is highly negatively correlated with the maximum blocking degree $S$, with a correlation coefficient of 0.9363, while the linear correlation coefficient for the dam breaching duration is only 0.1262. The test results show that the high flow velocity and discharge in the main channel and the low flow velocity and discharge in the branch channel are the key factors controlling the progressive river-blocking mode.

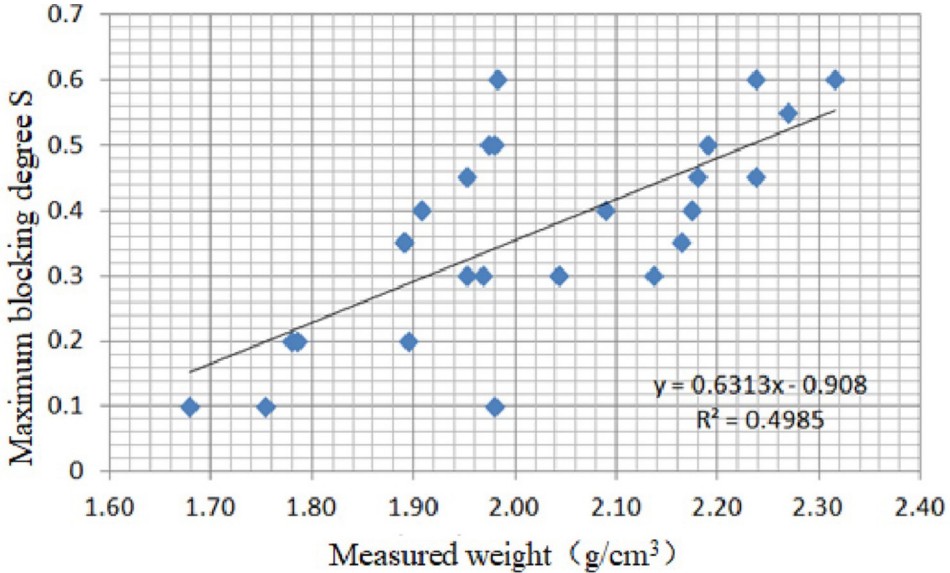

**Fig 10. Correlation between the measured density and the maximum blocking degree *S*.**

The discharge ratio was negatively correlated with the backwater height and the amount of washout, with poor correlations of 0.0039 and 0.0679, respectively. In this mode, a large proportion of the debris flow materials will remain on the opposite bank of the main river to form a lateral moraine, and this phenomenon is common in the study area.

## 5 Discussion

A series of field physical model tests were conducted in the middle reaches of the Dadu River, according to the test results, three typical river-blocking modes of debris flow are classified, i.e. thrust-type river blocking, mixed-flow-type river blocking and progressive river blocking. In spite of a total of 90 groups of physical tests were carried out in this study, the limitations of these field tests exist and the test conditions are hard to cover all the real disaster scenes in the study area, the test results need to be validated by more studies in other means, e.g. numerical simulations and theoretical analysis. In addition, the three typical river-blocking modes of debris flow proposed in this study are summarized based on the field tests in the study area, whether these modes are applicable to other regions remains to be tested.

The test results showed that the material composition is the the three typical river-blocking modes of debris flow. At present, the water flow velocity and water flow field [10–12], parameter sensitivity analysis [13–15], and scouring and sediment process analysis [16, 17] have attached the attention of many researchers, however, the effect of the material composition on the river-blocking mode is little studied. In practice, the river-blocking mode is determined by many factors, herein, only the main influence factor, i.e. the intersection angles, the longitudinal slope of the main channel and the branch channel, the slope of the branch channel, material composition, flow rate and discharge, were considered in this study, a detailed study considering more influence factors are needed, to form accurate river-blocking modes.

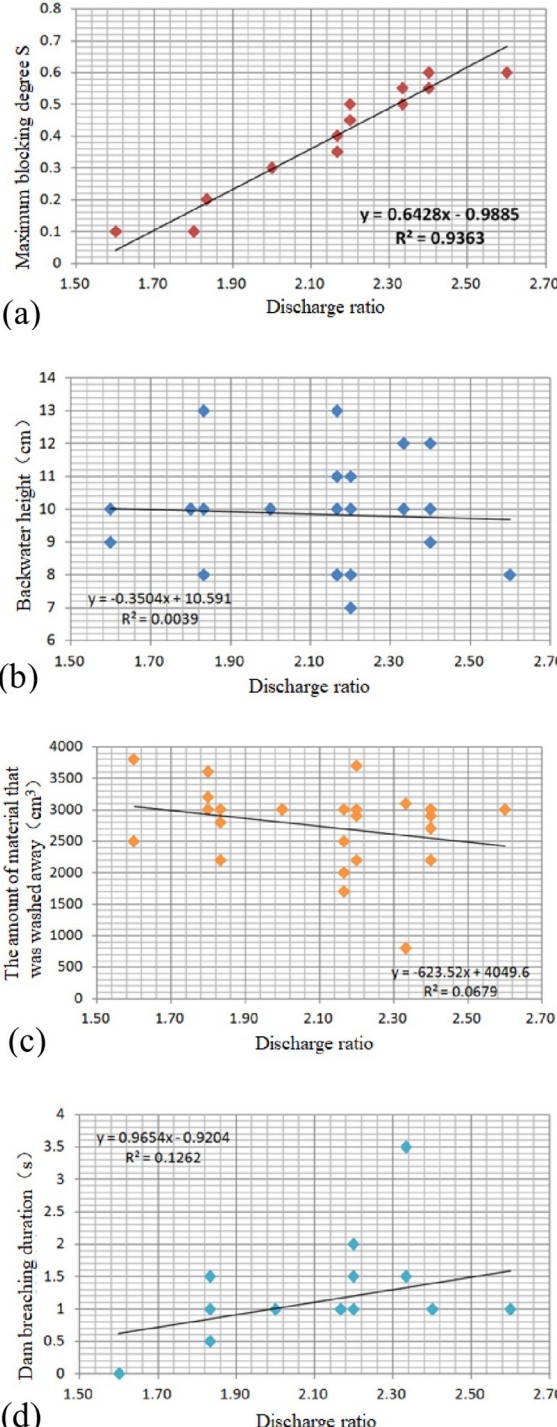

**Fig 11.** Correlation between discharge ratio and (a) the maximum blocking degree *S*, (b) backwater height, (c) the amount of material that was washed away and (d) dam breaching duration.

## 6 Conclusion

Field physical model tests were conducted in this study to reveal the mechanism of river blocking by debris flow in the middle reaches of the Dadu River, Southwest China. The follow conclusions can be drawn.

(1) Three typical river-blocking modes of debris flow can be summarized in these tests, i.e. thrust-type river blocking, mixed-flow-type river blocking and progressive river blocking.

(2) The material composition has a controlling effect on the thrust-type river-blocking model, only the tests that adopted the mixed soil and gravel exhibited thrust-type river blocking.

(3) The debris flows composed of sand tend to block the river with mixed-flow-type river-blocking mode, accounting for 50% of the occurrence in the model tests.

(4) The debris flows composed of sand is more likely to result in mixed-flow-type river blocking, the high flow rate and discharge in the main channel and the low flow rate and discharge in the branch channel are the favorable conditions for forming progressive river-blocking mode.

## Author Contributions

**Data curation:** Zhi Song, Yanni Chen, Gang Fan.

**Formal analysis:** Yanni Chen.

**Funding acquisition:** Gang Fan.

**Methodology:** Gang Fan.

**Writing – original draft:** Zhi Song, Yanni Chen.

**Writing – review & editing:** Yunxin Zhan.

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
