## [Decision Letter · Decision Letter 0]

1 Jun 2023

PONE-D-23-02409Field physical model tests on the mechanism of river blocking by debris flow in the middle reaches of the Dadu River, Southwest ChinaPLOS ONE

Dear Dr. Fan,

Thank you for submitting your manuscript to PLOS ONE. After careful consideration, we feel that it has merit but does not fully meet PLOS ONE’s publication criteria as it currently stands. Therefore, we invite you to submit a revised version of the manuscript that addresses the points raised during the review process.

We look forward to receiving your revised manuscript.

Kind regards,

Gowhar Meraj, Ph .D.

Academic Editor

PLOS ONE

“This research was financially supported by the Sichuan Province International Science and Technology Innovation Cooperation/ Hong Kong, Macao and Taiwan Science and Technology Innovation Cooperation Project (grant number 2021YFH0178, received by Gang Fan), the Open Fund of State Key Laboratory of Geohazard Prevention and Geoenvironment (No. SKLGP2021K008, received by Gang Fan).”

“This research was financially supported by the Sichuan Province International Science and Technology Innovation Cooperation/ Hong Kong, Macao and Taiwan Science and Technology Innovation Cooperation Project (grant number 2021YFH0178), the Open Fund of State Key Laboratory of Geohazard Prevention and Geoenvironment (No. SKLGP2021K008).”

“This research was financially supported by the Sichuan Province International Science and Technology Innovation Cooperation/ Hong Kong, Macao and Taiwan Science and Technology Innovation Cooperation Project (grant number 2021YFH0178, received by Gang Fan), the Open Fund of State Key Laboratory of Geohazard Prevention and Geoenvironment (No. SKLGP2021K008, received by Gang Fan).”

7. We note that [Figures 1-5, 8 and 12] in your submission contain [map/satellite] images which may be copyrighted. All PLOS content is published under the Creative Commons Attribution License (CC BY 4.0), which means that the manuscript, images, and Supporting Information files will be freely available online, and any third party is permitted to access, download, copy, distribute, and use these materials in any way, even commercially, with proper attribution. For these reasons, we cannot publish previously copyrighted maps or satellite images created using proprietary data, such as Google software (Google Maps, Street View, and Earth). For more information, see our copyright guidelines: http://journals.plos.org/plosone/s/licenses-and-copyright.

We require you to either (1) present written permission from the copyright holder to publish these figures specifically under the CC BY 4.0 license, or (2) remove the figures from your submission

A. You may seek permission from the original copyright holder of [Figures 1-5, 8 and 12] to publish the content specifically under the CC BY 4.0 license. 

B. If you are unable to obtain permission from the original copyright holder to publish these figures under the CC BY 4.0 license or if the copyright holder’s requirements are incompatible with the CC BY 4.0 license, please either i) remove the figure or ii) supply a replacement figure that complies with the CC BY 4.0 license. Please check copyright information on all replacement figures and update the figure caption with source information. If applicable, please specify in the figure caption text when a figure is similar but not identical to the original image and is therefore for illustrative purposes only.

Natural Earth (public domain): http://www.naturalearthdata.com/.

Additional Editor Comments:

Dear Dr. Gang Fan,

Thank you for submitting your manuscript to our journal. Your work has been carefully evaluated by two reviewers who have expertise in your research area. Based on their insightful comments and recommendations, I've decided that your manuscript requires major revisions before it can be considered for publication. Below are the key areas for improvement, as suggested by the reviewers:

1. Both reviewers noted issues with the language, readability, and organization of your manuscript. In particular, the abstract was seen as poorly organized and contained unclear expressions. I recommend that you seek the assistance of a professional editor to enhance the clarity, style, and structure of your manuscript.

2. There are concerns about the introduction, including a lack of critical review of relevant literature and an unclear statement of the research gap. Please revise this section to include a clear identification of the knowledge gaps your study intends to fill, based on an exhaustive review of related literature.

3. Your paper lacks a comprehensive discussion section. Both reviewers indicated that this section is crucial in an academic paper. Your revised manuscript should include a discussion of the achievements and limitations of your study, with perspectives on data, methods, and results. Additionally, this section should provide insight into future research or work prospects.

4. The reviewers also highlighted issues with the figures and tables included in your manuscript. Please revisit these and make necessary modifications to enhance their clarity and relevance.

5. Lastly, please revise your conclusion section to ensure it doesn't merely repeat the abstract. This section should succinctly summarize your main findings and their implications.

Please address each comment provided by the reviewers in your response letter and indicate the corresponding changes in your revised manuscript. These changes will greatly enhance the quality of your research and its presentation, improving its suitability for publication in our journal.

We look forward to receiving your revised manuscript and your detailed response to the reviewers' comments.

Best Regards,

Gowhar Meraj

Reviewers' comments:

Reviewer's Responses to Questions

**Comments to the Author**

1. Is the manuscript technically sound, and do the data support the conclusions?

Reviewer #1: Partly

Reviewer #2: Yes

2. Has the statistical analysis been performed appropriately and rigorously? 

Reviewer #1: No

Reviewer #2: Yes

3. Have the authors made all data underlying the findings in their manuscript fully available?

Reviewer #1: Yes

Reviewer #2: Yes

4. Is the manuscript presented in an intelligible fashion and written in standard English?

Reviewer #1: No

Reviewer #2: Yes

5. Review Comments to the Author

Reviewer #1: This interesting study investigated the dynamic processes and river-blocking modes of debris flow using field physical experiments. Although this is a very physical-intensive work, the paper is poorly written and lacks in-depth discussion.

1. Abstract is poorly organized with many wrong expressions and lacks a clear knowledge gap.

L24-25, a wrong sentence. L29-30, the logic is wrong.

L33-36, the implication is badly summarized.

2. This paper is poorly written that some paragraphs of the paper are in a lack of readability because of too redundant information and language issues. I suggest that the authors focus on the style/language and to ask a professional editor to go over the whole text.

3. Introduction would benefit from a critical review of relevant literature, identifying gaps in our knowledge and justifying their approach to the topic.

L45-L53, listing many historic stories is not good.

L57-69, these introductory content should appear in the Study area and Material.

L91-103, directly listing many references is very bad.

4. Figures 1-5 introduces the background of debris flow in the Dadu River, but this study focus on the field experiments. The relation between actual condition and experiment results should be separated. I think Figures 1-5 should be combined in one figures.

5. Figures 8b, 10b, 12b have low resolution.

6. In Figures 9, 11, 13, 14, fitting all data using the linear function is not appropriate.

7. Table 1 and Table 2 is useless.

8. Discussion is missing.

9. Conclusion almost repeats the Abstract.

Reviewer #2: This manuscript studies the mechanism of debris flow blocking the middle reaches of the Dadu River through field physical model experiments, revealing the dynamic process of debris flow blocking the river. Three types of debris flow blockage patterns were observed, providing support for understanding the mechanism of debris flow blockage and disaster prevention and reduction in the region. The specific suggestions are as follows: (1) Please add necessary references in some places, such as lines 48-53; (2) In the introduction, the summary and description of previous work, that is, the work of predecessors, lacks organization, indicating a lack of logic in the entire introduction. On the basis of summarizing previous work, deficiencies in previous research should be identified. Of course, this deficiency needs to be related to the work of this manuscript. This can lead to a more reasonable introduction to the work of this manuscript. Therefore, I suggest the author to reorganize the introduction. (3) This manuscript lacks a discussion section, which is the most important part of an academic paper. It is recommended that the author add it. The achievements and limitations of this study should be discussed from various aspects such as data, methods, and results. In addition, it is also necessary to provide several research or work prospects. (4) The language and expression of this manuscript are not very good, it is recommended to polish the manuscript. For example, in the abstract, the meaning of this sentence is not clear and needs to be modified –“The particle size and density of sand are more likely to block the river with mixed-flow-type river-blocking mode”.

In summary, my suggestion is a major revision.

6. PLOS authors have the option to publish the peer review history of their article (what does this mean?). If published, this will include your full peer review and any attached files.

Reviewer #1: No

Reviewer #2: **Yes: **Chong Xu

---

## [Author Response · Author response to Decision Letter 0]

9 Nov 2023

PLOS ONE

Response to the Comments from Reviewers

Manuscript number: PONE-D-23-02409

Full title: Field physical model tests on the mechanism of river blocking by debris flow in the middle reaches of the Dadu River, Southwest China

Article type: Research paper

Authors: Zhi Song, Yanni Chen, Gang Fan

We would like to thank the reviewer and the Editor for their comments and constructive suggestions. We have considered these comments and revised the manuscript accordingly. Listed below please find our responses to the editor’s and reviewer’ comments. 

Response to the Comments from reviewer #1

Q1. Abstract is poorly organized with many wrong expressions and lacks a clear knowledge gap. 

L24-25, a wrong sentence. L29-30, the logic is wrong. L33-36, the implication is badly summarized.

Response: Thanks for your comment. The abovementioned sentences are revised by the authors.

Q2. This paper is poorly written that some paragraphs of the paper are in a lack of readability because of too redundant information and language issues. I suggest that the authors focus on the style/language and to ask a professional editor to go over the whole text. 

Response: Thanks for your comment. The manuscript has been polished by a native English speaker. Attached is the editing certification of this manuscript.

Q3. Introduction would benefit from a critical review of relevant literature, identifying gaps in our knowledge and justifying their approach to the topic.

L45-L53, listing many historic stories is not good.

L57-69, these introductory content should appear in the Study area and Material.

L91-103, directly listing many references is very bad.

Response: Thanks for your comment. The manuscript has been revised. The introductory content about the middle reaches of the Dadu River has been moved to “the Study area and Material”. Additional, the citation of the references is revised.

Q4. Figures 1-5 introduces the background of debris flow in the Dadu River, but this study focus on the field experiments. The relation between actual condition and experiment results should be separated. I think Figures 1-5 should be combined in one figures.

Response: Thanks for your comment. As you mentioned, 5 figures are adopted in the manuscript to introduce the basis information of the study area, it seems superfluous. Accordingly, only 2 figures are retained to show the debris flows in the study area, i.e., “Figure 1 Typical debris flow cases in the study area” and “Figure 2 Distribution of debris flows and (b) debris flow density in the study area”, other figures are deleted. Since Figure 1 and Figure 2 include too many information, it is hard to merge these two figures into one figure, hence the Figure 1 and Figure 2 are retained in the revised manuscript.

Q5. Figures 8b, 10b, 12b have low resolution.

Response: Thanks for your comment. Figures 8b, 10b, 12b have been redrawn and improved.

Q6. In Figures 9, 11, 13, 14, fitting all data using the linear function is not appropriate.

Response: Thanks for your comment. According to the authors’ attempts, the linear function is the optimal one with the maximum correlation coefficient. Accordingly, a sentence is added in the revised manuscript to state this issue.

Q7. Table 1 and Table 2 are useless. 

Response: Thanks for your comment. Table 1 and Table 2 have been deleted in the manuscript, accordingly, some hydrological information of study area is added in the manuscript, please see Lines 71-73.

Q8. Discussion is missing.

Response: Thanks for your comment. A discussion is added in the revised manuscript.

Q9. Conclusion almost repeats the Abstract.

Response: Thanks for your comment. The conclusion is revised.

Response to the Comments from reviewer #2

Q1. Please add necessary references in some places, such as lines 48-53. 

Response: Thanks for your comment. The citation of the references in the manuscript is revised.

Q2. In the introduction, the summary and description of previous work, that is, the work of predecessors, lacks organization, indicating a lack of logic in the entire introduction. On the basis of summarizing previous work, deficiencies in previous research should be identified. Of course, this deficiency needs to be related to the work of this manuscript. This can lead to a more reasonable introduction to the work of this manuscript. Therefore, I suggest the author to reorganize the introduction. 

Response: Thanks for your comment. The introduction has been revised according to the comments. The revised manuscript has emphasized the research gap between the current studies and this study.

Q3. This manuscript lacks a discussion section, which is the most important part of an academic paper. It is recommended that the author add it. The achievements and limitations of this study should be discussed from various aspects such as data, methods, and results. In addition, it is also necessary to provide several research or work prospects. 

Response: Thanks for your comment. A discussion is added in the revised manuscript to discuss the data, method and the research results, and also the potential prospects on the future research.

Q4. The language and expression of this manuscript are not very good, it is recommended to polish the manuscript. For example, in the abstract, the meaning of this sentence is not clear and needs to be modified –“The particle size and density of sand are more likely to block the river with mixed-flow-type river-blocking mode”.

Response: Thanks for your comment. The whole manuscript was checked and polished carefully.

---

## [Decision Letter · Decision Letter 1]

2 Jan 2024

PONE-D-23-02409R1Field physical model tests on the mechanism of river blocking by debris flow in the middle reaches of the Dadu River, Southwest ChinaPLOS ONE

Dear Dr. Fan,

Thank you for submitting your manuscript to PLOS ONE. After careful consideration, we feel that it has merit but does not fully meet PLOS ONE’s publication criteria as it currently stands. Therefore, we invite you to submit a revised version of the manuscript that addresses the points raised during the review process.

We look forward to receiving your revised manuscript.

Kind regards,

Gowhar Meraj, Ph .D.

Academic Editor

PLOS ONE

Journal Requirements:

Additional Editor Comments:

The authors are suggested to provide a rigorous discussion supported by an indepth literature survey.

Reviewers' comments:

Reviewer's Responses to Questions

**Comments to the Author**

1. If the authors have adequately addressed your comments raised in a previous round of review and you feel that this manuscript is now acceptable for publication, you may indicate that here to bypass the “Comments to the Author” section, enter your conflict of interest statement in the “Confidential to Editor” section, and submit your "Accept" recommendation.

Reviewer #1: All comments have been addressed

Reviewer #2: (No Response)

2. Is the manuscript technically sound, and do the data support the conclusions?

Reviewer #1: Yes

Reviewer #2: Yes

3. Has the statistical analysis been performed appropriately and rigorously? 

Reviewer #1: Yes

Reviewer #2: Yes

4. Have the authors made all data underlying the findings in their manuscript fully available?

Reviewer #1: Yes

Reviewer #2: Yes

5. Is the manuscript presented in an intelligible fashion and written in standard English?

Reviewer #1: Yes

Reviewer #2: Yes

6. Review Comments to the Author

Reviewer #1: This study carried out field physical model tests to study the mechanism of river blocking by debris flow in the middle reaches of the Dadu River. It tried to reveal the dynamic processes of river blocking by debris flows and proposed three typical river-blocking modes of debris flow. By reviewing the revised manuscript, all the comments from previous reviewers have been fully responded and the manuscript has been revised properly. This revised manuscript could be reconsidered for publication with minor revisions. The comments on this version are as follows.

(1) The research gap is suggested to enhance in the manuscript, to address the significant importance to the disaster prevention and mitigation in the middle section of the Dadu River.

(2) Determining of geometric scale is a key factor in the design of physical model tests, so the authors are suggested to state the reason why the geometric scale was determined to be 1: 500 and its effect on the test results.

(3) The key factors that determine the specific river-blocking modes of debris flow should be discussed in the Discussion.

Reviewer #2: The author has made revisions to the manuscript, resulting in an improvement in its quality. Regarding the discussion section, however, the current discussion section is too shallow and it is difficult to imagine the discussion section of an academic paper without references, which is unacceptable. Suggest the authors rewrite the discussion section. The achievements and limitations of this study should be discussed from various aspects such as data, methods, and results In addition, it is also necessary to provide multiple research or work perspectives.

7. PLOS authors have the option to publish the peer review history of their article (what does this mean?). If published, this will include your full peer review and any attached files.

Reviewer #1: **Yes: **Zhiwei Li

Reviewer #2: No

---

## [Author Response · Author response to Decision Letter 1]

13 Jan 2024

PLOS ONE

Response to the Comments from Reviewers

Manuscript number: PONE-D-23-02409

Full title: Field physical model tests on the mechanism of river blocking by debris flow in the middle reaches of the Dadu River, Southwest China

Article type: Research paper

Authors: Zhi Song, Yunxin Zhan, Yanni Chen, Gang Fan

We would like to thank the reviewer and the Editor for their comments and constructive suggestions. We have considered these comments and revised the manuscript accordingly. Listed below please find our responses to the editor’s and reviewer’ comments. 

Response to the Comments from reviewer #1

Q1. The research gap is suggested to enhance in the manuscript, to address the significant importance to the disaster prevention and mitigation in the middle section of the Dadu River.

Response: Thanks for your comment. The significant importance of this study has been addressed in the revised manuscript, please see Lines 107-111.

Q2. Determining of geometric scale is a key factor in the design of physical model tests, so the authors are suggested to state the reason why the geometric scale was determined to be 1: 500 and its effect on the test results.

Response: Thanks for your comment. According to the limitations of the field test site and the capacity of the test equipment, the geometric scale of this model test was finally determined to be 1:500. Accordingly, the reason why the geometric scale was determined to be 1: 500 is stated in the revised manuscript. Please see Lines 241-246.

Q3. The key factors that determine the specific river-blocking modes of debris flow should be discussed in the Discussion.

Response: Thanks for your comment. The discussion was revised, please see Lines 519-538.

Response to the Comments from reviewer #2

Q1. The author has made revisions to the manuscript, resulting in an improvement in its quality. Regarding the discussion section, however, the current discussion section is too shallow and it is difficult to imagine the discussion section of an academic paper without references, which is unacceptable. Suggest the authors rewrite the discussion section. The achievements and limitations of this study should be discussed from various aspects such as data, methods, and results In addition, it is also necessary to provide multiple research or work perspectives. 

Response: Thanks for your comment. The discussion was revised, please see Lines 519-538.

---

## [Editor Report · Decision Letter 2]

18 Jan 2024

Field physical model tests on the mechanism of river blocking by debris flow in the middle reaches of the Dadu River, Southwest China

PONE-D-23-02409R2

Dear Dr. Fan,

We’re pleased to inform you that your manuscript has been judged scientifically suitable for publication and will be formally accepted for publication once it meets all outstanding technical requirements.

Kind regards,

Gowhar Meraj, Ph .D.

Academic Editor

PLOS ONE

Additional Editor Comments (optional):

Thanks